# Two level logistic regression analysis of factors influencing skilled birth attendant during delivery among Bangladeshi mothers: A nationally representative sample

**Nusrat Jahan Ema[1], Mahmuda Khanom Eva[1], Abu Sayed Md. Al Mamun[1], Md. Abdur Rafi[2], Ahsanul Khabir[3], Ramendra Nath Kundu[4], Premananda Bharati[5], Md. Golam Hossain**[1] *

1 Health Research Group, Department of Statistics, University of Rajshahi, Rajshahi, Bangladesh,
2 Rajshahi Medical College, Rajshahi, Bangladesh, 3 Medical Officer, Medical Centre, University of Rajshahi, Rajshahi, Bangladesh, 4 Research Associate-I, Indian Council of Medical Research—Centre for Ageing & Mental Health, Kolkata, West Bengal, India, 5 Biological Anthropology Unit, Indian Statistical Institute, Kolkata, West Bengal, India

* hossain95@yahoo.com

## Abstract

### Background

In order to minimize the maternal and child mortality rate, the presence of skilled birth attendants (SBA) during delivery is essential. By 2022, 4th health, population and nutrition sector programme in Bangladesh aims to increase the percentage of deliveries performed by SBA to 65 percent. The objective of the present study was to determine the rate and associated factors of usage SBA among Bangladeshi mothers during their delivery.

### Methods

This study utilized secondary data that was collected by Bangladesh Demographic and Health Survey (BDHS) 2017–18. The usage of SBA was measured by a question to respondent, who assisted during your delivery? It was classified into two classes; (i) skilled birth attendant (qualified doctors, nurses, midwives, or paramedics; family welfare visitors, community skilled birth attendants, and sub-assistant community medical officers) (code 1), and (ii) unskilled birth attendant (untrained traditional birth attendants, trained traditional birth attendants, relatives, friends, or others) (code 0). Two logistic regression model was used to determine the associated factors of SBA after removing the cluster effect of the outcome variable.

### Results

This study found 53.2% mothers were delivered by SBA in Bangladesh, among them 56.33% and 42.24% mothers were delivered by nurse/midwife/paramedic and doctor respectively. The two level logistic model demonstrated that geographical location (division), type of residence, religion, wealth index, mothers' body mass index, mothers' education level, mothers' occupation, total ever born children, mothers' age at first birth (year), number

**Data Availability Statement:** Data are freely available at https://dhsprogram.com/data/dataset/Bangladesh_Standard-DHS_2017.cfm?flag=0.

**Funding:** The authors received no specific funding for this work.

**Competing interests:** The authors have declared that no competing interests exist.

**Abbreviations:** ANC, Antenatal care; AOR, Adjusted odds ratio; BBS, Bangladesh Bureau of Statistics; BDHS, Bangladesh Demographic and Health Survey; CI, Confidence interval; IBM, International business machines corporation; IC, individual contribution; EAs, Enumeration areas; HPNSP, Health population and nutrition sector program; MMR, maternal mortality rate; MOR, median odds ratio; NPV, Positive predictive value; PPV, Positive predictive value; ROC, receiver operator characteristic; SBA, Skilled birth attendants; SDGs, Sustainable development goals; TBAs, Traditional birth attendants; SE, standard error.

of ANC visits, husbands' education level and husbands' occupation were significant (p<0.01) predictors of SBA. Mothers' education and wealth index were the most important contributory factors for SBA in Bangladesh.

## Conclusions

This study revealed that still 46.8% mothers are delivered by unskilled birth attendant, this might be treated of Bangladesh Government to achieve SDGs indicator 3.1.2 by 2030. Counseling could be integrated during ANC to increase awareness, and should ensure for every Bangladeshi mothers visit ANC service during their pregnancy at least 4 times.

## Introduction

A skilled attendant is an accredited health professional, such as a midwife, doctor or nurse—who has been educated and trained to proficiency in the skills needed to manage normal pregnancies, childbirth and the immediate postnatal period, and in the identification, management and referral of complications in women and newborns [1–3]. Obstetric care from a trained provider during delivery is a critical factor in reducing maternal and neonatal mortality; these are the major issues in many countries, especially in developing countries like Bangladesh where health and medically related reforms are being actively implemented [3, 4].

Globally about 295 thousand women died during and following pregnancy and childbirth in 2017 and approximate neonatal deaths occurred in 2018 was 2.5 million [5]. The UN inter-agency estimated that, from 2000 to 2017, global maternal mortality declined by 38 percent, from 342 deaths to 211 deaths per 100,000 live births. Maternal mortality ratio in South Asia is 163, which is 19 percent of the global total [6]. Between 2003 and 2017, maternal mortality ratio of Bangladesh was declining at a moderating rate to from 395 deaths per 100,000 live births in 2003 to 173 deaths per 100,000 live births in 2017. Globally, nearly 300,000 women die annually from preventable causes around the time of childbirth which is more than one maternal death in every two minutes.

In Bangladesh, every year loses 7,660 women to preventable causes related to pregnancy and childbirth. Only 4% of all births in Bangladesh are attended by a skilled provider, with 71 percent of all births taking place at home, and only 36% of women receive any pregnancy care from a skilled provider [7]. 49% of births were delivered in a health facility in 2018 (32 percent in private facilities, 14% in public facilities, and 4% in nongovernmental organization facilities). The number of facilities delivered increased by 12 percentage points between 2014 and 2017–18 [8]. Only 7% of non-institutional deliveries followed the five recommended essential newborn care practices. In 2020, 2.4 million children died in their first month of life worldwide. About 6,500 neonatal deaths every day, which is about two deaths every minute. Nearly one third of all neonatal deaths are occurring within the first day of life and nearly three-quarters occurring within the first week [6]. Bangladesh's neonatal mortality rate was 17.5 deaths per 1,000 live births in 2020. Bangladesh's neonatal mortality rate dropped from 96.3 deaths per 1,000 live births in 1971 to 17.5 deaths per 1,000 live births in 2020. Only a small percentage (7%) of newborns received the five recommended essential newborn care practices (using a safe delivery kit or boiled blade to cut the umbilical cord, applying nothing or only chlorhexidine to the cord, immediate drying after birth, bathing delayed 72 hours or more, and immediate breastfeeding). The fourth HPNSP's goal is to increase coverage of essential newborn care practices from 6% in 2014 to 25% by 2022 [9]. By 2022, the 4th HPNSP aims to raise the

percentage of deliveries performed by medically trained providers, also known as skilled birth attendants to 65 percent [9]. Slightly more than half of all deliveries (53%) were attended by medically trained providers. Untrained traditional birth attendants (TBAs) assisted 35% of births, trained TBAs 10%, and relatives, friends, or others 2%. The percentage of births delivered by a medically trained provider has steadily increased over time, rising from 21% in 2007 to 32% in 2011, 42% in 2014, and 53% in 2017–18. The rapid rise in deliveries by medically trained providers has been fueled by an increase in facility deliveries. The proportion of non-institutional deliveries by medically trained providers remained at 5% or less between 2007 and 2017–18 [8]. The likelihood of giving birth in a hospital has risen over time. Facility deliveries increased from 4% to 49% between 1993–94 and 2017–18, owing primarily to the rapid increase in private sector deliveries since 2007. The percentage of women who give birth in a private facility has risen from 8% in 2007 to 22% in 2014 and 32% in 2017–18. In the case of public facilities, it increased from 7% in 2007 to 13% in 2014, and then to 14% in 2017–18 [8].

Bangladesh's government is committed to meeting the targets for Sustainable Development Goals 2 and 3 (SDGs 2 and 3). Bangladesh wants to reduce the MMR to 121 deaths per 100,000 live births, the under-5 mortality rate to 34 deaths per 1,000 live births, and the neonatal mortality rate to 18 deaths per 1,000 live births by 2022, according to the 4th Health, Population, and Nutrition Sector Program (4th HPNSP) [10]. Maternal mortality is the third leading cause of death in Bangladesh, accounting for 14% of all deaths among women aged 15 to 49. High-quality prenatal care and skilled labor and delivery attendance could prevent the majority of these deaths, including obstetric hemorrhage (which accounts for nearly one-third of maternal deaths) and eclampsia (which accounts for 20% of maternal deaths) [11]. Complications and infections that can result in death or serious illness for the mother or the newborn can be reduced with proper medical attention and hygienic conditions during delivery. As a result, increasing the proportion of births performed in a safe, clean environment under the supervision of health professionals is critical [12, 13]. The Bangladesh Maternal Health Strategy, which encourages women to give birth in the presence of medically trained birth attendants, promotes safe motherhood through a variety of activities, including skilled birth attendant (SBA) delivery [14]. The majority of maternal deaths can be avoided because there are well-established health-care solutions to prevent or manage complications [15]. A skilled health professional (doctor, nurse, or midwife) during delivery is crucial in reducing maternal and child deaths. It has been proven that usage skilled birth attendants (SBA) during and after childbirth reduces maternal and newborn deaths [16, 17]. Maternal and newborn health is inextricably linked. All births should be attended by skilled health professionals, because prompt management and treatment can mean the difference between life and death for both the mother and the baby [18].

Researchers in Bangladesh should pay special attention to mothers and newborn health for achieving SDGs by 2030. It is essential to investigate current situation of skilled birth attendants' facilities in the country. Some researchers determined the associated factors of skilled birth attendants during delivery of mothers, most of researches did research with locally population not nationally used much smaller data sets that were not representative of the nation [19–24]. Some studies used nationally representative samples [22, 24], data came from the different levels, there were more probable to get cluster effect of outcome variable without removing the effect may get misleading results. However, one study used multilevel logistic regression to overcome the effect. To the best of our knowledge, no researchers did try to determine the contribution of individual associated factors of skilled birth attendants which are very important for considering factors according their importance to improve health policy for increasing skilled birth attendants of a particular population [24].

Therefore, researchers in the present study wanted to determine associated factors of skilled birth attendant during the delivery of Bangladeshi mothers using two level logistic regression model, in addition, the study tried to calculate the contribution of individual associated factors.

## Methods

### Unite level data

In this study, a total of 5012 Bangladeshi mothers were extracted from BDHS, 2017–18, they selected sample for collecting data from across the country; it was a nationally representative samples. The age of mothers was 15–49 years. The main objective of the survey was to provide up-to-date information on fertility and fertility preferences; childhood mortality levels and causes of death; awareness, approval, and use of family planning methods; maternal and child health, including breastfeeding practices and nutritional status; newborn care; women's empowerment; selected non-communicable diseases; and availability and accessibility of health and family planning services at the community level [8]. The cross-sectional data were collected between October 24, 2017 and March 15, 2018. The survey collected all possible information to fulfill their objectives. The survey design, sampling technique, survey instruments, pre and post tested of questionnaire, measuring system and quality control have been described elsewhere [8].

### Inclusion criteria

We considered women in their reproductive age in Bangladesh who gave at least one birth in the 3 years before the survey, and did not have any serious diseases.

### Sampling procedure

BDHS, 2017–2018 used two-stage stratified sampling for selecting samples from the population. In the first step, 675 enumeration areas (EAs) created by Bangladesh Bureau of Statistics (BBS) were chosen with a probability proportional to EA size (250 in urban areas and 425 in rural areas) [25]. A systematic sample of 30 families per EA was selected in the second stage of sampling to provide statistically reliable estimates of key demographic and health characteristics for the country as a whole, for urban and rural areas separately, and for each of the eight divisions. A total of 20,250 residential homes were chosen based on this concept, and finally, BDHS, 2017–2018 selected 20,127 ever married women aged 15–49 years for the survey. We found some missing values in the data, and these values were excluded. Then after removing mothers who did not give birth in three years preceding the BDHS 2017–18 surveys, our data set was reduced to 5012 for the analysis of this study. The Fig 1 illustrates the sample selection process for this study.

### Outcome variable

The outcome variable of the study was status of birth attendant during delivery of Bangladeshi mothers, it was classified into two classes; (i) skilled birth attendant (SBA) (qualified doctors, nurses, midwives, or paramedics; family welfare visitors, community skilled birth attendants, and sub-assistant community medical officers) (code = 1), and (ii) unskilled birth attendant (untrained traditional birth attendants, trained traditional birth attendants, relatives, friends, or others) (code = 0).

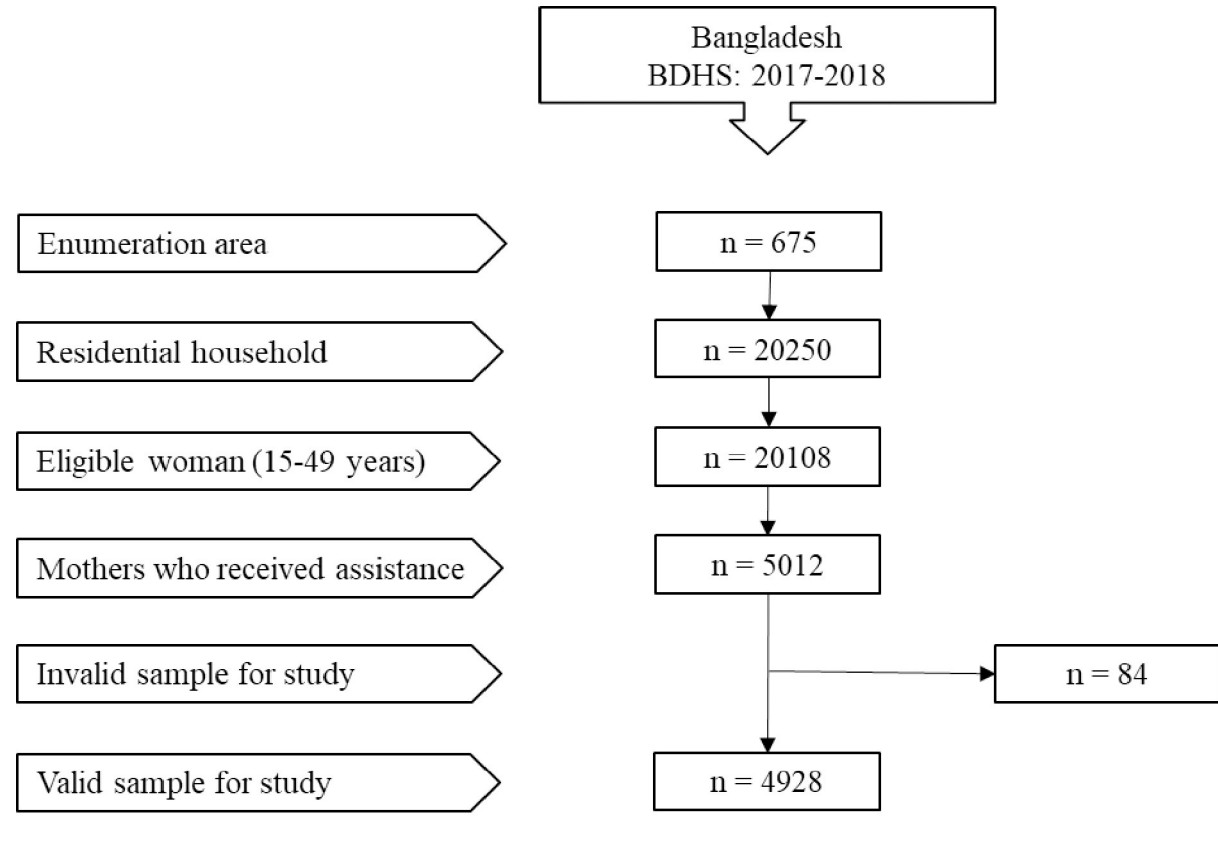

**Fig 1. Sampling procedure for the present study.**

### Independent variable

In this study, independent variables included socioeconomic, demographic, household information and anthropometric factors. All selected independent variables with their groups are mentioned in Table 1.

### Ethics approval and consent to participate

2017–2018 BDHS received ethics approval from the Ministry of Health and Family Welfare, Bangladesh respectively. The survey also received written consent from each individual in the study.

### Statistical analysis

BDHS, 2017–2018 mentioned that any analysis using their data set required application of sampling weights to ensure the actual representation of the survey results at the national and division levels. The 2017–18 BDHS sampling weights were not expected to lead to any significant differences in the overall survey indicators [8]. Frequency distribution (percentage) was used to determine the percentage of usage of skilled birth attendant during delivery of

**Table 1. General characteristics of samples and the status of birth attendant by the characteristics.**

| Independent variables | | Status of birth attendant | |
|---|---|---|---|
| | | Unskilled, N (%) | Skilled, N (%) |
| Geographical location (Division) | | | |
| | Barisal, 527(10.7) | 264 (50.1) | 263 (49.9) |
| | Chittagong, 809(16.4) | 394 (48.7) | 415 (51.3) |
| | Dhaka, 712(14.4) | 272 (38.2) | 440 (61.8) |
| | Khulna, 515(10.5) | 176 (34.2) | 339 (65.8) |
| | Mymensingh, 601(12.2) | 333 (55.4) | 268 (44.6) |
| | Rajshahi, 523(10.6) | 224 (42.8) | 299 (57.2) |
| | Rangpur, 558(11.3) | 263 (47.1) | 295 (52.9) |
| | Sylhet, 683 (13.9) | 381 (55.8) | 302 (44.2) |
| Type of residence | | | |
| | Urban, 1676(34.0) | 566 (33.8) | 1110 (66.2) |
| | Rural, 3252(66.0) | 1741 (53.5) | 1511(46.5) |
| Religion | | | |
| | Muslim, 4509(91.5) | 2153 (47.7) | 2356 (52.3) |
| | Non-Muslim, 419(8.5) | 154 (36.8) | 265 (63.2) |
| Number of family member | | | |
| | ≤4, 1497(30.4) | 665 (44.4) | 832 (55.6) |
| | 5 to 6, 1799(36.5) | 866 (48.1) | 933 (51.9) |
| | ≥7, 1632(33.1) | 776 (47.5) | 856 (52.5) |
| Type of toilet facility | | | |
| | Hygienic, 2904(58.9) | 1140 (39.3) | 1764 (60.7) |
| | Unhygienic, 2024(41.1) | 1167 (57.7) | 857 (42.3) |
| Wealth index | | | |
| | Poor, 2075(42.1) | 1368 (65.9) | 707 (34.1) |
| | Middle, 898(18.2) | 420 (46.8) | 478 (53.2) |
| | Rich, 1955(39.7) | 519 (26.5) | 1436(73.5) |
| Relationship to household head | | | |
| | Wife, 2612(53.0) | 1332 (51.0) | 1280 (49.0) |
| | Other, 2316(47.0) | 975 (42.1) | 1341(57.9) |
| Mothers' age group (year) | | | |
| | ≤20, 1222(24.8) | 571 (46.7) | 651 (53.3) |
| | 21–30, 2880(58.4) | 1329 (46.1) | 1551(53.9) |
| | 31–40, 787(16.0) | 390 (49.6) | 397 (50.4) |
| | 41–49, 39(0.8) | 17 (43.6) | 22 (56.4) |
| Mothers' body mass index | | | |
| | Underweight, 791(16.1) | 463(58.5) | 328(41.5) |
| | Normal, 3000(60.9) | 1512(50.4) | 1488(49.6) |
| | Overweight, 1137(23.1) | 332(29.2) | 805(70.8) |
| Mothers' education level | | | |
| | No education, 307(6.2) | 222 (72.3) | 85 (27.7) |
| | Primary, 1378(28.0) | 908 (65.9) | 470 (34.1) |
| | Secondary, 2360(47.9) | 1028 (43.6) | 1332(56.4) |
| | Higher, 883(17.9) | 149 (16.9) | 734 (83.1) |
| Mothers' occupation | | | |
| | Housewife, 2963(60.1) | 1225 (41.3) | 1738 (58.7) |
| | Others, 1965(39.9) | 1082 (55.1) | 883 (44.9) |

(*Continued*)

**Table 1.** (Continued)

| Independent variables | | Status of birth attendant | |
|---|---|---|---|
| | | Unskilled, N (%) | Skilled, N (%) |
| Current marital status | | | |
| | Living with husband, 4862(98.7) | 2274 (46.8) | 2588 (53.2) |
| | Others, 66(1.3) | 33 (50.0) | 33(50.0) |
| Total ever born children | | | |
| | 1 child, 1883(38.2) | 653 (34.7) | 1230 (65.3) |
| | 2 child, 1611(32.7) | 775 (48.1) | 836 (51.9) |
| | $\geq$3 child, 1434(29.1) | 879 (61.3) | 555 (38.7) |
| Mothers' age at first birth (year) | | | |
| | $\leq$ 20, 3868(78.5) | 2000 (51.7) | 1868 (48.3) |
| | > 20, 1060(21.5) | 307 (29.0) | 753 (71.0) |
| Mothers' age at first marriage (year) | | | |
| | $\leq$18, 4040(82.0) | 2058 (50.9) | 1982 (49.1) |
| | >18, 888(18.0) | 249 (28.0) | 639 (72.0) |
| Number of ANC visits | | | |
| | Inadequate (<4 times), 2155(43.7) | 1235(57.3) | 920(42.7) |
| | Adequate ($\geq$4 times), 2773(56.3) | 1072(38.7) | 1701(61.3) |
| Place of delivery | | | |
| | Home, 2446(49.6) | 2287(93.5) | 159(6.5) |
| | Others, 2482(50.4) | 20(0.8) | 2462(99.2) |
| Husbands' education level | | | |
| | No education, 685(13.9) | 472(68.9) | 213(31.1) |
| | Primary, 1637(33.2) | 960(58.6) | 677(41.4) |
| | Secondary, 1608(32.6) | 674(41.9) | 934(58.1) |
| | Higher, 998(20.3) | 201(20.1) | 797(79.9) |
| Husbands' occupation | | | |
| | Unemployed, 119(2.4) | 56(47.1) | 63(52.9) |
| | Farmer, 916(18.6) | 585(63.9) | 331(36.1) |
| | Hard work, 2546(51.7) | 1228(48.2) | 1318(51.8) |
| | Service holder, 315(6.4) | 38(12.1) | 277(87.9) |
| | Business, 1032(20.9) | 400(38.8) | 632(61.2) |

Bangladeshi mothers. Both univariate (unadjusted) and multivariate (adjusted) binary logistic models were used to find the effect of socio-economic, demographic and other selected variables on skilled birth attendant. All variables with a p-value of <0.20 from the likelihood ratio test in the univariate model were included in the multivariate logistic regression model. The secondary data was collected data using two-stage stratified cluster sampling. The data came from several levels of hierarchy; it was possible to obtain a clustering effect in outcome variable. The traditional model (single-level statistical model) would not be appropriate for analyzing the hierarchical data if cluster effect exists [26]. Two levels of multiple logistic regression analysis were used to remove the clustering effect for finding the effect of selected independent variables on the status of birth attendant during delivery of Bangladeshi mothers. The multilevel logistic regression model is a powerful statistical tool for detecting associated factors of outcome variable after removing its cluster effect [27]. We used median odds ratio (MOR) for checking the existence of clustering effect in outcome variable. The MOR is defined as:
MOR = $\exp\{0.6745 \sqrt{2}\sqrt{\sigma_u^2}\} = \exp(0.95\sqrt{\sigma_u^2})$, where $\sigma_u^2$ is the cluster variance. The value of MOR is always greater than or equal to 1, and if MOR = 1, means there is no cluster variation,

but cluster effect exists if MOR>1, and it should be removed otherwise get misleading results [27]. We calculated the value of MOR, which quantifies the variation among clusters, and the MOR value was 2.16 which indicated the multilevel model was appropriate for analyzing the data. The magnitude of the standard error (SE) was used to detect the multicollinearity problem among the independent variables, if the SE lies between 0.001 and 0.5, it is judged that there is no evidence of multicollinearity [28]. The fitness of both selected models was tested using Hosmer and Lemeshow test. The model's accuracy was tested by sensitivity, specificity, positive and negative predictive values and receiver operator characteristic (ROC) curve. Finally, we used stepwise logistic regression analysis for selecting the most influential predictors for skilled birth attendants of Bangladeshi mothers, and calculated the corresponding $R^2$ change for determining the individual contribution (IC) of the risk factors for the outcome variable using by the formula, IC = (individual $R^2$ change/total $R^2$)×100% [29]. Statistical significance was accepted at $p<0.05$. Statistical analyses were carried out using STATA (version 13) and SPSS software (version IBM 25).

## Results

A total number of 4928 married women were included in this study to determine the rate of usage of skilled birth attendant among Bangladeshi mothers. It determined the rate of usage skilled birth attendant and investigated the effect of socio-economic, demographic factors and anthropometric factors on usage of skilled birth attendant among Bangladeshi mothers.

It was found that 53.20% mothers were delivered by skilled birth attendant, among them more than half of the mothers (56.33%) by nurse/midwife/paramedic and 42.24% by doctor. A few mothers were delivered by family welfare visitor (0.77%), community skilled birth attendant (0.55%) and community health care provider (0.08%) (Fig 2).

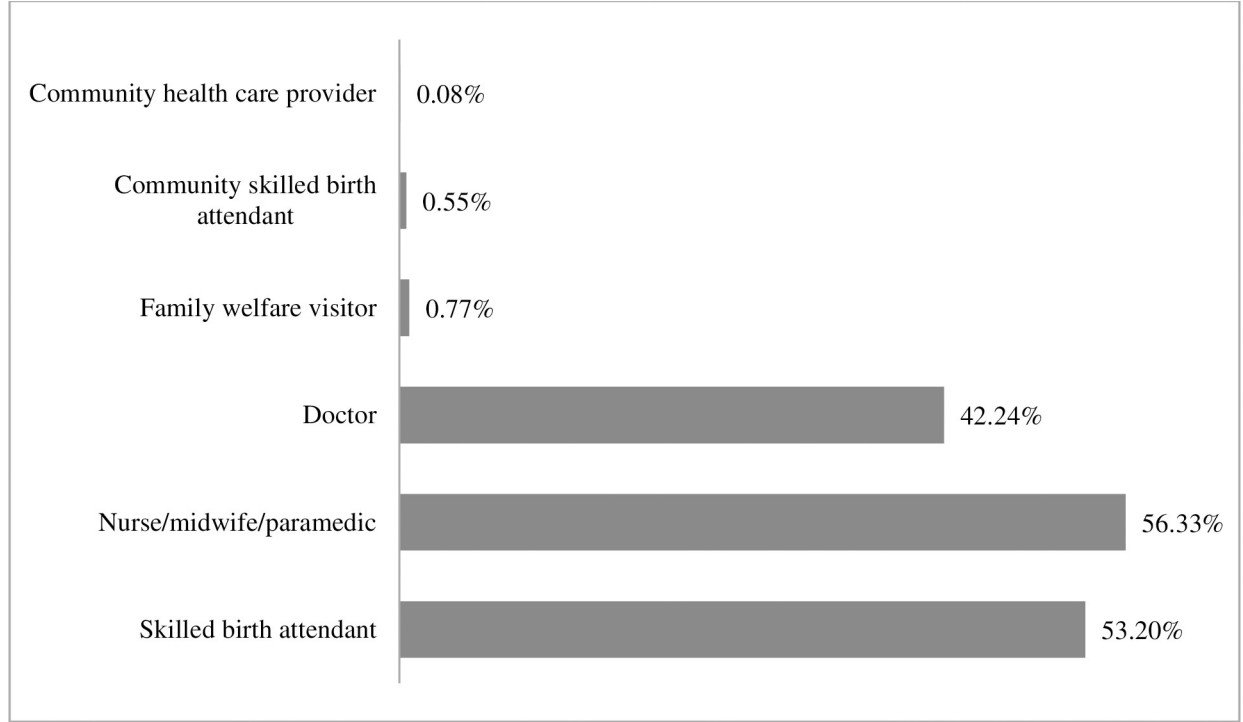

**Fig 2. Rate of skilled birth attendant delivered Bangladeshi mothers.**

We observed that the highest number of mothers (65.8%) living in Khulna division was delivered by skilled birth attendant while lowest in Sylhet division (44.2%). 66.0% mothers were selected from rural area, and more number of urban mothers (66.2%) was delivered by skilled birth attendant compared to rural mothers (46.5%). More than 90% mothers were Muslim, and higher number of non-Muslim mothers was delivered by skilled birth attendant than Muslim mothers. There was no major variation of mothers delivered by skilled birth attendant among family size. More number of mothers was delivered by skilled birth attendant who had hygienic toilet facility at their household compared to mothers did not have hygienic toilet facility. It was found increased the number of mothers delivered by skilled birth attendant with increasing their household wealth index. There was no major variation of mothers delivered by skilled birth attendant among age groups. We found increased the number of mothers delivered by skilled birth attendant with increasing their body mass index. Also, it was found that increased the number of mothers delivered by skilled birth attendant with increasing education level. Higher number of mothers delivered by skilled birth attendant who were currently living with their husbands compared to their counterpart. Number of mothers delivered by skilled birth attendant was decreasing with increasing their number of total ever born children. Lower number of early married (age≤18) and early child bearing mothers (age at first birth ≤20 years) were delivered by skilled birth attendant than their counterpart. More number of mothers delivered by skilled birth attendant who received adequate (≥4 times) ANC during their pregnancy period compared to their counterpart. Only 6.5% mothers were delivered by skilled birth attendant who delivered at home. We found increased the number of mothers delivered by skilled birth attendant with increasing their husband education level. The highest number of service holder husbands' wife (87.9) delivered by skilled birth attendant among all husbands' occupation groups (Table 1).

## Multilevel logistic regression model

Table 2 shows the unadjusted odds ratio (OR) and adjusted odds ratio (AOR) of skilled birth attendant by different background characteristics among mothers in Bangladesh. As we mentioned earlier, the independent variables for multivariate multilevel logistic model were selected on the basis of p-value (p-value<0.2) from univariate model, and we interpreted the results came from multivariate model. SE demonstrated that there was no evidence of multicollinearity problems among independent variables in multivariate model (Table 2). After controlling the effect of other factors, the model showed that mothers living in Khulna (AOR 2.12; CI 1.47, 3.06; p<0.01), Rajshahi (AOR 1.72; CI 1.19, 2.47; p<0.01) and Rangpur (AOR 1.59; CI 1.10, 2.30; p<0.05) divisions were more likely to use SBA than those living in Sylhet. Mothers living in urban areas had 28% higher chance to use SBA than rural mothers (AOR 0.72; CI 0.58, 0.88; p<0.01). Women with higher education had 63%, 59% and 39% more chance to use SBA than mothers with no (AOR 0.37; CI 0.24, 0.56; p<0.01), primary (AOR 0.41; CI 0.31, 0.55; p<0.01) and secondary (AOR 0.64; CI 0.49, 0.82; p<0.01) education. Mothers having 1 child were more likely to use SBA than those who have ≥3 children (AOR 1.98; CI 1.63, 2.41; p<0.01). Mothers who got first child at greater than 20 years were more likely to use SBA than those who got first child at ≤ 20 years (AOR 1.37; CI 1.08, 1.74; p< 0.01). Non-housewife mothers had 25% more chance to use SBA than housewife mothers (AOR 0.75; CI 0.64, 0.87; p<0.01). Mothers living in rich family had 55% and 39% more chance to use SBA than those living in poor (AOR 0.45; CI 0.36, 0.56; p<0.01) and middle (AOR 0.61; CI 0.501, 0.76; p<0.01) families respectively. Overweight women were more likely to use SBA than normal weighted women (AOR 1.83; CI 1.52, 2.19; p<0.01). Mothers whose husbands were doing businessman were 26% more likely to use SBA than farmer wives (AOR 0.74; CI 0.59, 0.94; p<

**Table 2. Effects of demographic, anthropometric, socioeconomic and behaviors factors on use of skilled birth attendant using multilevel logistic regression.**

| Independent variables | Unadjusted OR (95% CI: Lower-Upper) | Adjusted OR (95% CI: Lower-Upper) |
|---|---|---|
| Geographical location (Division) | p-value<0.2 | |
| Barisal vs Sylhet[R] | 1.242(0.987–1.564) | 1.33 (0.92, 1.92) |
| Chittagong vs Sylhet[R] | 1.334(1.086–1.639)** | 1.06 (0.76, 1.48) |
| Dhaka vs Sylhet[R] | 2.055(1.659–2.546)** | 1.37 (0.97, 1.93) |
| Khulna vs Sylhet[R] | 2.458(1.936–3.120)** | 2.12 (1.47, 3.06)** |
| Mymensingh vs Sylhet[R] | 0.993(0.794–1.242) | 1.17 (0.82, 1.67) |
| Rajshahi vs Sylhet[R] | 1.645(1.305–2.074)** | 1.72 (1.20, 2.47)** |
| Rangpur vs Sylhet[R] | 1.404(1.119–1.761)** | 1.60 (1.11, 2.30)* |
| Type of residence | p-value<0.2 | |
| Rural vs Urban[R] | 2.271(2.009–2.568)** | 0.72 (0.59, 0.88)** |
| Religion | p-value<0.2 | |
| Non-Muslim vs Muslim[R] | 1.511(1.225–1.864)** | 1.49 (1.12, 1.98)** |
| Type of toilet facility | p-value<0.2 | |
| Unhygienic vs Hygienic[R] | 0.475(0.423–0.533)** | 0.91 (0.77, 1.07) |
| Wealth index | p-value<0.2 | |
| Poor vs Rich[R] | 0.187(0.163–0.214)** | 0.46 (0.37, 0.57)** |
| Middle vs Rich[R] | 0.411(0.349–0.485)** | 0.62 (0.50, 0.76)** |
| Relationship to household head | p-value<0.2 | |
| Others vs Wife[R] | 1.431(1.279–1.602)** | 1.03 (0.88, 1.21) |
| Mothers' age group (year) | p-value>0.2 | |
| ≤20 vs 41-49[R] | 0.881(0.463–1.675) | |
| 21–30 vs 41-49[R] | 0.902(0.477–1.705) | |
| 31–40 vs 41-49[R] | 0.787(0.411–1.504) | |
| Mothers' body mass index | p-value<0.2 | |
| Underweight vs Normal Weight[R] | 0.720(0.614–0.844)** | 0.85 (0.70, 1.03) |
| Overweight vs Normal Weight[R] | 2.464(2.128–2.853)** | 1.83 (1.52, 2.20)** |
| Mothers' education level | p-value<0.2 | |
| No education vs Higher[R] | 0.078(0.057–0.106)** | 0.37 (0.25, 0.56)** |
| Primary vs Higher[R] | 0.105(0.085–0.129)** | 0.42 (0.31, 0.56)** |
| Secondary vs Higher[R] | 0.263(0.217–0.319)** | 0.64 (0.50, 0.82)** |
| Mothers' occupation | p-value<0.2 | |
| Others vs Housewife[R] | 0.575(0.513–0.645)** | 0.75 (0.64, 0.88)** |
| Currently marital status | p-value>0.2 | |
| Living with husband vs others[R] | 1.138(0.700–1.850) | |
| Total ever born children | p-value<0.2 | |
| 1 children vs ≥3 children[R] | 2.983(2.587–3.440)** | 1.99 (1.64, 2.41)** |
| 2 child vs ≥3 children[R] | 1.708(1.479–1.974)** | 1.18 (0.98, 1.42) |
| Mothers' age at first birth (year) | p-value<0.2 | |
| > 20-year vs ≤ 20 years[R] | 2.626(2.267–3.042)** | 1.37 (1.08, 1.74)** |
| Mothers' age at first marriage (year) | p-value<0.2 | |
| >18 years vs ≤18 years[R] | 2.665(2.273–3.124)** | 1.07 (0.82, 1.40) |
| Number of ANC visits | p-value<0.2 | |
| Adequate (≥4times) vs Inadequate(<4 times) [R] | 2.130(1.899–2.389)** | 1.61 (1.39, 1.85)** |
| Husbands' education level | p-value<0.2 | |
| 2.130(1.899–2.389)** | 0.098(0.78–0.124)** | 0.49 (0.36, 0.68)** |

(*Continued*)

**Table 2.** (Continued)

| Independent variables | Unadjusted OR (95% CI: Lower-Upper) | Adjusted OR (95% CI: Lower-Upper) |
|---|---|---|
| Primary vs Higher[®] | 0.154(0.126–0.186)** | 0.54 (0.41, 0.71)** |
| Secondary vs Higher[®] | 0.302(0.248–0.366)** | 0.64 (0.49, 0.82)** |
| Husbands' occupation | p-value<0.2 | |
| Unemployed vs Business[®] | 0.914(0.495–1.689) | 1.12 (0.54, 2.36) |
| Farmer vs Business[®] | 0.358(0.298–0.430)** | 0.74 (0.59, 0.94)* |
| Hard work vs Business[®] | 0.679(0.586–0.787)** | 0.91 (0.76, 1.09) |
| Service holder vs Business[®] | 4.614(3.214–6.622)** | 1.41 (0.90, 2.21) |

Hosmer and Lemeshow test, $\chi^2$ value = 12.51 (p-value 0.13)

**N.B.:** CI = Confidence Interval, OR = Odds ratio

® = Reference category

0.05). Non-Muslim mothers had 1.48 fold higher chances to use SBA than Muslim mothers (AOR 1.48; CI 1.11, 1.98; p< 0.01). Mothers having adequate ANC ($\geq$4 times) visits were more likely to use SBA than inadequate ANC visits (<4 times) (AOR 1.60; CI 1.39, 1.85; p<0.01). Higher educated husbands' wife were 51% 46% and 37% more likely to get SBA during their delivery than no (AOR 0.49; CI 0.35, 0.68; p<0.01), primary (AOR 0.54; CI 0.411, 0.71; p< 0.01) and secondary (AOR 0.63; CI 0.49, 0.82; p<0.01) husbands' wife. Hosmer and Lemeshow test showed that our selected model was good fitted (p>0.05) (Table 2).

## ROC curve for use on SBA

ROC was used to measure the accuracy of the test. The accuracy was measured by the area under the ROC curve, and the area of the ROC curve was 0.785, by which we could say that 78.5% of all possible pairs of subjects in which one used SBA and one had not use SBA. This model would assign a higher probability to the subject who used skilled birth attendant (Fig 3).

## Classification of subjects

The overall accuracy of this model to predict the usage of skilled birth attendant (with a predicted probability of 0.5 or greater) was 71.3% (Table 3). The sensitivity is given by 1585/2269 = 69.85% and the specificity is 1876/2584 = 72.6%. Positive predictive value (PPV) 1585/2293 = 69.12% and negative predictive value (NPV) 1876/2560 = 73.28% (Table 3).

## Stepwise logistic regression analysis

The final step of the stepwise logistic regression is shown in Table 4. In the model, all variables were included in the first step, for the next step, the variable was excluded based on the Wald statistic value (least significant) and insignificant change in −2LR. In the second step, mothers' age at first marriage was least significant and the change in −2LR was insignificant; therefore, this variable was excluded from the model in the third step. In the third step, type of toilet facility was least significant and the change in −2LR was insignificant; therefore, this variable was excluded from the model in the fourth step. The final step of the model included division, type of residence, mothers' educational level, total children ever born, mothers' age at first birth, mothers' occupation, wealth index, mothers' BMI, husbands' occupation, religion, number of ANC visits and husbands' education level, this was the best subset model for status of birth attendant in Bangladesh. It was observed that household wealth quintile and mothers'

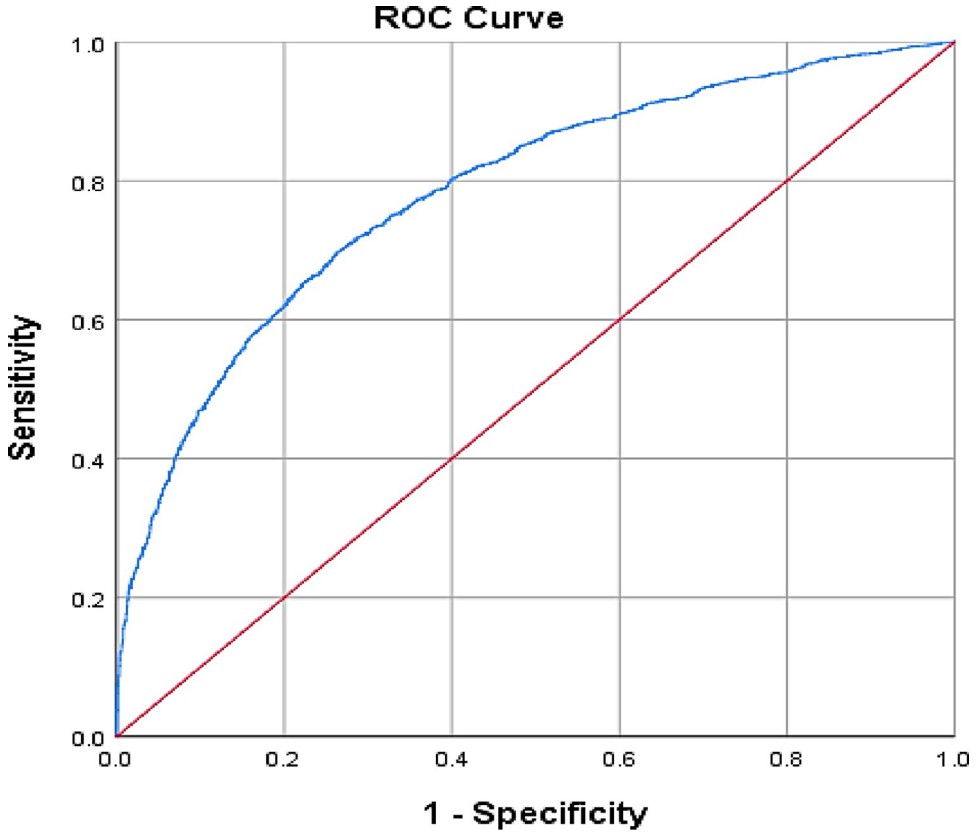

**Fig 3. ROC curve and area under usage of skilled birth attendant.**

education put highest contribution (18.74%) amongst all major predictors for getting skilled birth attendants while religion was found as lowest contributory predictor (Table 4).

## Discussion

In this study, we attempted to investigate the present situation in status of birth attendant during delivery of Bangladeshi mother using nationally representative sample. Some studies had been done with Bangladeshi population but most of them used small sample size collected in a regional though two studies used nationally representative sample but they did not appropriate statistical models [19–24]. We used multilevel logistic model for removing cluster effect of outcome variable with Hosmer and Lemeshow test and ROC curve for justifying the model. In addition, model discrimination was used in this study to show the classification of subject.

**Table 3. Classification of subject by binary logistic regression analysis for usage of SBA.**

| Skilled Birth Attendant | | Predicted | | Percentage Correct |
|---|---|---|---|---|
| | | Unskilled | Skilled | |
| Observed | Unskilled | 1585 | 684 | 69.9 |
| | Skilled | 708 | 1876 | 72.6 |
| Overall Percentage | | | | 71.3 |

**Table 4. Best subset model and individual contribution for skilled birth attendants.**

| *Final step* | | | | | | | | | | |
|---|---|---|---|---|---|---|---|---|---|---|
| | Independent variables | B | SE | Wald | p-value | AOR | 95% CI of AOR | | Change in | Individual |
| | | | | | | | Lower | Upper | -2 LR | contribution (%) |
| Geographical location (Division) | | | | | | | | | 39.27 | 3.09 |
| | Barisal vs Sylhet[R] | 0.27 | 0.14 | 3.82 | 0.051 | 1.31 | 1.00 | 1.72 | | |
| | Chittagong vs Sylhet[R] | 0.05 | 0.12 | 0.14 | 0.713 | 1.05 | 0.82 | 1.33 | | |
| | Dhaka vs Sylhet[R] | 0.29 | 0.13 | 5.07 | 0.024 | 1.34 | 1.04 | 1.72 | | |
| | Khulna vs Sylhet[R] | 0.68 | 0.14 | 22.90 | 0.001 | 1.97 | 1.49 | 2.60 | | |
| | Mymensingh vs Sylhet[R] | 0.15 | 0.13 | 1.18 | 0.278 | 1.16 | 0.89 | 1.51 | | |
| | Rajshahi vs Sylhet[R] | 0.52 | 0.14 | 13.88 | 0.001 | 1.68 | 1.28 | 2.20 | | |
| | Rangpur vs Sylhet[R] | 0.41 | 0.14 | 8.77 | 0.003 | 1.51 | 1.15 | 1.98 | | |
| Type of residence | | | | | | | | | 10.70 | 5.37 |
| | Rural vs Urban[R] | 0.26 | 0.08 | 10.74 | 0.001 | 1.29 | 1.11 | 1.51 | | |
| Religion | | | | | | | | | 11.12 | 0.57 |
| | Non-Muslim vs Muslim[R] | -0.41 | 0.12 | 10.99 | 0.001 | 0.66 | 0.52 | 0.85 | | |
| Wealth index | | | | | | | | | 89.13 | 18.74 |
| | Poor vs Rich[R] | -0.86 | 0.09 | 88.37 | 0.001 | 0.42 | 0.35 | 0.51 | | |
| | Middle vs Rich[R] | -0.47 | 0.10 | 23.15 | 0.001 | 0.63 | 0.52 | 0.76 | | |
| Mothers' body mass index | | | | | | | | | 55.96 | 6.29 |
| | Underweight vs Normal Weight[R] | -0.18 | 0.09 | 3.52 | 0.061 | 0.84 | 0.70 | 1.01 | | |
| | Overweight vs Normal Weight[R] | 0.58 | 0.09 | 45.17 | 0.000 | 1.79 | 1.51 | 2.13 | | |
| Mothers' education level | | | | | | | | | 58.60 | 18.74 |
| | No education vs Higher[R] | -1.05 | 0.19 | 30.30 | 0.001 | 0.35 | 0.24 | 0.51 | | |
| | Primary vs Higher[R] | -0.89 | 0.14 | 43.21 | 0.001 | 0.41 | 0.31 | 0.53 | | |
| | Secondary vs Higher[R] | -0.43 | 0.12 | 13.11 | 0.001 | 0.65 | 0.51 | 0.82 | | |
| Mothers' occupation | | | | | | | | | 16.98 | 2.74 |
| | Others vs Housewife[R] | 0.30 | 0.07 | 17.00 | 0.001 | 1.35 | 1.17 | 1.55 | | |
| Total ever born children | | | | | | | | | 57.70 | 7.09 |
| | 1 children vs ≥3 children[R] | 0.61 | 0.09 | 47.49 | 0.001 | 1.84 | 1.55 | 2.19 | | |
| | 2 child vs ≥3 children[R] | 0.12 | 0.09 | 1.91 | 0.167 | 1.13 | 0.95 | 1.34 | | |
| Mothers' age at first birth (year) | | | | | | | | | 15.06 | 5.37 |
| | > 20-year vs ≤ 20 years[R] | -0.35 | 0.09 | 14.97 | 0.001 | 0.70 | 0.59 | 0.84 | | |
| Number of ANC visits | | | | | | | | | 49.79 | |
| | Adequate (≥4times) vs Inadequate(<4 times)[R] | -0.48 | 0.07 | 49.76 | 0.001 | 0.62 | 0.55 | 0.71 | | |
| Husbands' education level | | | | | | | | | 19.90 | 5.14 |
| | No education vs Higher[R] | -0.62 | 0.15 | 16.43 | 0.001 | 0.54 | 0.40 | 0.73 | | |
| | Primary vs Higher[R] | -0.53 | 0.13 | 16.75 | 0.001 | 0.59 | 0.46 | 0.76 | | |
| | Secondary vs Higher[R] | -0.39 | 0.12 | 9.85 | 0.002 | 0.68 | 0.53 | 0.87 | | |
| Husbands' occupation | | | | | | | | | 12.07 | 9.49 |
| | Unemployed vs Business[R] | 0.16 | 0.36 | 0.21 | 0.649 | 1.18 | 0.58 | 2.38 | | |
| | Farmer vs Business[R] | -0.30 | 0.11 | 7.39 | 0.007 | 0.74 | 0.60 | 0.92 | | |
| | Hard work vs Business[R] | -0.12 | 0.09 | 1.88 | 0.170 | 0.89 | 0.75 | 1.05 | | |
| | Service holder vs Business[R] | 0.32 | 0.22 | 2.21 | 0.137 | 1.38 | 0.90 | 2.11 | | |
| | Constant | 1.62 | 0.22 | 55.57 | 0.000 | 5.07 | | | | |

**N.B.:** AOR = Adjusted odds ratio, CI = Confidence Interval, SE = Standard Error, B = Co-efficient

[R] = Reference category.

Stepwise logistic was used to find the best subset model, and also individual contribution of major predictors were calculated.

## Percentage of usage of skilled birth attendant

We found the rate of usage of skilled birth attendant is more 53.2%. The proportion of births delivered by a SBA provider has been increasing in Bangladesh with increasing women educational qualification, household wealth quintile and number of ANC visit over the time [8]. However, the proportion of SBA some countries in Asia such as in Nepal (62%), Pakistan (69%), India (81%), Myanmar (60%), Thailand (99%), China (100%), Japan (100%), South Korea (100%), Malaysia (100%), Afghanistan (59%), Indonesia (95%), Papua New Guinea (56%), Philippines (84%) [30]. The proportion of births delivered by a SBA provider was low compared to some Asian countries because health care facility is not available to many Bangladeshi women, to overcome the problem Bangladesh Government more focused on providing proper health care facilities of maternal health [8].

## Associated factors of usage of skilled birth attendant

We found that the variation of using SKB were significant among geographical location (division) it may be occurred due to different custom and culture and socio-economic status [31]. Same results had been found previous study in Bangladesh [22, 32]. We found that more number of urban mothers uses SKB during their delivery than rural mothers, same findings had been found in other studies [22, 24, 32]. Mothers living in urban environment are more educated and living in family having higher wealth quintile than rural are [8], and educated and rich mothers are more conscious about their health comparatively less educated and poor mothers. Also medical facilities in urban area are more than rural in Bangladesh; consequently urban mothers are more likely to get SKB than rural mothers. We also observed that comparatively higher educated and rich mothers were more likely to use SKB than their counterparts; these results are supported to the findings of other studies [21, 22, 24, 32]. Education is an important factor for using SBA during delivery. A Nigerian study demonstrated that women with advanced education were multiple times more likely to use SBA than less educated women [33]. Rich mothers were more likely to use SBA than poor and middle-class women. Wealth index is an important factor for using SBA in Bangladesh; health care service is more affordable for rich people, these results agreed with the previous findings [21, 22, 24, 32]. Similar findings were seen in Afghanistan, where an increase in the wealth index enhanced the rate of utilization of safe delivery care among the mothers [34]. In southeast Ethiopia, the utilization of trained birth attendants was associated with residence, mother education, and ANC [12]. Economic disparity has an impact on trained birth attendants in India, yet there is a tendency toward improvement [17, 31]. Women who had 3 or more than 3 children were less likely to use SBA than who had only one child. The reason might be age. Women tend to have some complexity during the birth of their first child. A study of Ghana found that women with four or more births had the highest proportion of skilled deliveries in terms of parity [35], which disagreed with our result, but our result supported to findings of other studies in Bangladesh [22, 32]. Women whose age were >20 years during their first birth were more likely to use SBA than who was younger during their first birth. This might be an effect of lack of self-consciousness, which increases with age. Housewife mothers were more likely to use SBA than others. Overweight women were more likely to use SBA than normal weight women, because normally overweight women belong to rich family. Mothers whose husband is farmer are less likely to use SBA than businessman husband. Muslim mothers were less likely to use SBA than others, same findings had been found in Nigerian study, they found that Muslim women were

less likely to use SBA than Christian women [33]. Mothers who had adequate ANC visits were more likely to use SBA than who had inadequate ANC visits. Proper ANC visit raises awareness among mothers so they tend to take proper delivery service. Same findings had been found in a study in Ghana, they found that the extent of women who got ANC and used skilled delivery administrations was high (91.5%) [36], also our findings supported to other Bangladesh studies [21, 22, 24, 32]. The coexistence of the ANC and SBA was observed in the Sub-Saharan Africa, and these were mostly related to economic class and geographic clusters of countries [37]. Husbands' education level is an important factor for using SBA. Mothers having higher educated husband were more likely to use SBA than who had less educational qualification. Same findings had been found in other studies [21, 22, 24, 32].

## Conclusions

In our present study we considered 4928 Bangladeshi mothers aged 15–49 as a sample to investigate the factors influencing the usage of SBA during their delivery. This cross-sectional study revealed that 53.2% mothers used SBA during their delivery. Two level logistic model provided that geographical location (division), type of residence, mothers and their husbands' education level, total number of ever born children, mothers' age at first birth, mothers and their husbands' occupation, household wealth quintile, mothers' BMI, religion and ANC are significantly associated factors of SBA. Among these factors, mothers' education and household wealth quintile are the most contributory factors for SBA. These factors could be considered for increasing to use SBA during delivery of Bangladeshi mothers in order to rise in order to raise skilled birth attendants to 65% as the aim of 4th HPNSP in Bangladesh for achieving SGDs indicator 3.1.2 by 2030 [9]. This study may also help to reduce the global maternal mortality ratio. The entire world is presently moving to the fourth industrial revolution, with the goal of making the rhetoric of "health care for all" a reality.

## Strength and limitation of the study

The current study looked at the characteristics influencing not usage skilled birth attendant during delivery of Bangladesh mothers using multilevel logistic regression model. The study carried out using recent nationally representative data. However, we had some limitation, it was a cross-sectional study, it was unable to determine causality in the connections between the covariates and the outcome variables, and therefore we could only draw probabilistic conclusions. Secondary data was used in this study, could not possible to consider other possible important predictors of SBA. Clearly more research is required regarding SBA among Bangladeshi mothers.

## Acknowledgments

The authors would like to acknowledge Bangladesh Demographic and Health Survey (BDHS-2017-2018) for providing their data.

## Author Contributions

**Conceptualization:** Nusrat Jahan Ema, Abu Sayed Md. Al Mamun, Md. Golam Hossain.

**Data curation:** Mahmuda Khanom Eva, Abu Sayed Md. Al Mamun, Ahsanul Khabir.

**Formal analysis:** Nusrat Jahan Ema, Mahmuda Khanom Eva, Abu Sayed Md. Al Mamun, Ramendra Nath Kundu, Md. Golam Hossain.

**Investigation:** Premananda Bharati, Md. Golam Hossain.

**Methodology:** Md. Abdur Rafi, Ahsanul Khabir, Md. Golam Hossain.

**Resources:** Md. Golam Hossain.

**Supervision:** Md. Golam Hossain.

**Validation:** Md. Abdur Rafi, Premananda Bharati, Md. Golam Hossain.

**Visualization:** Premananda Bharati.

**Writing – original draft:** Nusrat Jahan Ema, Mahmuda Khanom Eva, Ramendra Nath Kundu.

**Writing – review & editing:** Mahmuda Khanom Eva, Abu Sayed Md. Al Mamun, Md. Abdur Rafi, Ahsanul Khabir, Ramendra Nath Kundu, Premananda Bharati, Md. Golam Hossain.

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
