## [Decision Letter · Decision Letter 0]

11 May 2023

PONE-D-22-27776Two level logistic regression analysis of factors influencing skilled birth attendant during delivery among Bangladeshi mothers:  A nationally representative samplePLOS ONE

Dear Dr. Hossain,

Thank you for submitting your manuscript to PLOS ONE. After careful consideration, we feel that it has merit but does not fully meet PLOS ONE’s publication criteria as it currently stands. Therefore, we invite you to submit a revised version of the manuscript that addresses the points raised during the review process.

We look forward to receiving your revised manuscript.

Kind regards,

Mpho Keetile, PhD

Academic Editor

PLOS ONE

Journal Requirements:

Additional Editor Comments:

Please address the comments of the reviewers and your article will be considered for publication

Reviewers' comments:

Reviewer's Responses to Questions

**Comments to the Author**

1. Is the manuscript technically sound, and do the data support the conclusions?

Reviewer #1: Yes

2. Has the statistical analysis been performed appropriately and rigorously? 

Reviewer #1: Yes

3. Have the authors made all data underlying the findings in their manuscript fully available?

Reviewer #1: Yes

4. Is the manuscript presented in an intelligible fashion and written in standard English?

Reviewer #1: Yes

5. Review Comments to the Author

Reviewer #1: Comments to Authors

1. I strongly encourage that instead of chi-square values, please consider presenting an estimate of unadjusted relative risk (or odds) and confidence intervals. This will enable separation of the factors into those associated with an increase in risk for skilled birth attendant during delivery and those associated with a decreased risk.

2. In this study the two level logistic model was used to analysis the factors influencing the skilled birth attendant during delivery but ICC (intra class correlation) values not present in the report. Thus, please include ICC report to identify the total variation in skilled birth attendant during delivery due to regions.

6. PLOS authors have the option to publish the peer review history of their article (what does this mean?). If published, this will include your full peer review and any attached files.

Reviewer #1: No

---

## [Author Response · Author response to Decision Letter 0]

24 May 2023

Date: May 24, 2023

Response to reviewers 

Paper Title: Two level logistic regression analysis of factors influencing skilled birth attendant during delivery among Bangladeshi mothers: A nationally representative sample

Journal Name: PLOS ONE

Manuscript ID: PONE-D-22-27776

Dear Editor

Thank you very much for providing us valuable comments on our manuscript. We have modified and revised the manuscript accordingly; detailed point-by-point corrections are given below:

Journal Requirements:

Response: Thank you very much for providing Journal requirements. We have checked our manuscript, and we tried to follow PLOS ONE's style throughout the manuscript. 

Response: Thank you. 

Response: Since we did not receive any funding for this study, we have stated “The authors received no specific funding for this work.”

Response: We have checked, and the reference list and citations are ok. 

Reviewers' comments to the Author

1. Is the manuscript technically sound, and do the data support the conclusions?

Reviewer #1: Yes

2. Has the statistical analysis been performed appropriately and rigorously?

Reviewer #1: Yes

3. Have the authors made all data underlying the findings in their manuscript fully available?

Reviewer #1: Yes

4. Is the manuscript presented in an intelligible fashion and written in standard English?

Reviewer #1: Yes

Response: Thank you very much for your comments on our manuscript. 

5. Review Comments to the Author

Reviewer #1: Comments to Authors

1. I strongly encourage that instead of chi-square values, please consider presenting an estimate of unadjusted relative risk (or odds) and confidence intervals. This will enable separation of the factors into those associated with an increase in risk for skilled birth attendant during delivery and those associated with a decreased risk.

Response: We appreciate reviewer’s good comments. As reviewer’s comments, we have used both univariate (unadjusted) and multivariate (adjusted) binary logistic models to find the effect of socio-economic, demographic and other selected variables on skilled birth attendant. All variables with a p-value of <0.20 from the likelihood ratio test in the univariate model were included in the multivariate logistic regression model. We estimated unadjusted odds risk with its 95% confidence intervals instead of Chi-square test. Please see in Table 2. 

2. In this study the two level logistic model was used to analysis the factors influencing the skilled birth attendant during delivery but ICC (intra class correlation) values not present in the report. Thus, please include ICC report to identify the total variation in skilled birth attendant during delivery due to regions.

Response: We appreciate reviewer’s important comment. Reviewer is right, the intra class correlation (ICC) should use to check whether multiple level model is necessary or not for hierarchical data. We know, Larsen and Merlo (2005) suggest using the median odds ratio (MOR) for dichotomies outcome instead of ICC but still we know some researchers use ICC. We think ICC or MOR is suitable for checking whether multiple level model is necessary or not, and MOR has been used in many publications such as (i) Islam M.A, Mamun A, Hossain M.M, Bharati P, Saw A, Lestrel PE, et al. (2019) Prevalence and factors associated with early initiation of breastfeeding among Bangladeshi mothers: A nationwide cross-sectional study. PLoS ONE 14(4): e0215733. https://doi.org/10.1371/ journal.pone.0215733. 

In our present study, we have used MOR, we have mentioned the formula with the properties of MOR, and the value of MOR has been mentioned in page: 11; Line: 5-12. 

We would like to thank the reviewers for the valuable comments. We have revised the documents to the best of our ability, but we will definitely be happy to provide further improvement if there are further clarifications required. 

With best regards

Dr. Md. Golam Hossain

Professor of Health Research Group

Department of Statistics, University of Rajshahi

Rajshahi-6205, Bangladesh

E-mail: hossain95@yahoo.com

---

## [Decision Letter · Decision Letter 1]

6 Sep 2023

Two level logistic regression analysis of factors influencing skilled birth attendant during delivery among Bangladeshi mothers:  A nationally representative sample

PONE-D-22-27776R1

Dear Dr. Hossain,

We’re pleased to inform you that your manuscript has been judged scientifically suitable for publication and will be formally accepted for publication once it meets all outstanding technical requirements.

Kind regards,

Felix Chikaike Clement Wekere

Academic Editor

PLOS ONE

Additional Editor Comments (optional):

Thank you for revising your manuscript and addressing review comments. Kindly proofread the manuscript and address  the following points below to improve the manuscript.

Page 2 , Abstract, -*By 2022, 4th health, population and nutrition sector programme in Bangladesh aims to increase the percentage of deliveries performed by SBA to 65 percent*. Kindly cross check the year and revise accordingly.

Page 2, under result, ….. *The two level logistic model provided the significant.... edit the sentence*

Page 15, line 8, *Mothers living in rich family had 55% and 39% had more chance to use SBA than those living in poor ……..*     Kindly delete ‘had’ after 39%, and add ‘respectively’ at the end of the sentence.

Page 18, line 3, *…….. the least significant variable based on the Wald statistic was relationship to household head and also in −2LR was insignificant….* Kindly proofread this sentence and make correction.

Page 23, line 2 ….   *but our result is supported to other studies in Bangladesh [22, 32].* Kindly correct grammar for ease of flow.

Reviewers' comments:

Reviewer's Responses to Questions

**Comments to the Author**

1. If the authors have adequately addressed your comments raised in a previous round of review and you feel that this manuscript is now acceptable for publication, you may indicate that here to bypass the “Comments to the Author” section, enter your conflict of interest statement in the “Confidential to Editor” section, and submit your "Accept" recommendation.

Reviewer #1: All comments have been addressed

2. Is the manuscript technically sound, and do the data support the conclusions?

Reviewer #1: Yes

3. Has the statistical analysis been performed appropriately and rigorously? 

Reviewer #1: Yes

4. Have the authors made all data underlying the findings in their manuscript fully available?

Reviewer #1: Yes

5. Is the manuscript presented in an intelligible fashion and written in standard English?

Reviewer #1: Yes

6. Review Comments to the Author

Reviewer #1: Thank you very much for your responses.

You have tried to address all comments. Hence, I have decided to recommend to accept your manuscript.

7. PLOS authors have the option to publish the peer review history of their article (what does this mean?). If published, this will include your full peer review and any attached files.

Reviewer #1: No

---

## [Editor Report · Acceptance letter]

18 Sep 2023

PONE-D-22-27776R1 

Two level logistic regression analysis of factors influencing skilled birth attendant during delivery among Bangladeshi mothers:  A nationally representative sample 

Dear Dr. Hossain:

I'm pleased to inform you that your manuscript has been deemed suitable for publication in PLOS ONE. Congratulations! Your manuscript is now with our production department. 

Kind regards, 

on behalf of

Dr. Felix Chikaike Clement Wekere 

Academic Editor

PLOS ONE